# Tumour-Associated MUC1 Exerts Multiple Effects on Cholesterol and Lipid Metabolism—A Potential Pathogenic Effector of Atherosclerosis in Cancer

**DOI:** 10.3390/ijms27010518

**Published:** 2026-01-04

**Authors:** Yunliang Chen, Michael Scully

**Affiliations:** Thrombosis Research Institute, London SW3 6LR, UK

**Keywords:** tumour-associated MUC1 (TA-MUC1), atherosclerosis, cancer, cholesterol and lipid metabolism, foam cells

## Abstract

(1) Cancer has been shown to contribute to the progression of atherosclerosis, while inflammatory aspects of atherosclerosis can exert profound effects on cancer development and outcomes. TA-MUC1 (Tumour-associated Mucin 1) is a transmembrane glycoprotein that is overexpressed in many human epithelial cancers lining the intestine. Interestingly, the lack of intestinal MUC1 has been shown to impair cholesterol uptake in MUC1^−/−^ mice. (2) To investigate whether TA-MUC1 could have specific effects on cholesterol metabolism and, thereby, have the potential of impacting the pathogenesis of atherosclerosis in cancer patients. (3) The effect of TA-MUC1 on cholesterol and lipid metabolism was assayed using MUC1 gene knock down breast cancer cells. An in vitro coculturing model similar to in vivo biological conditions was used to determine that TA-MUC1 could also modulate the cholesterol metabolism of other cells. (4) Reduction or inhibition of TA-MUC1 activity resulted in a significant alteration in a number of the signalling pathways and proteins that are relevant to abnormal cholesterol metabolism (*p* < 0.0001). Coculturing of TA-MUC1 cancer cells with THP-1 cells also notably effectively induced monocytic THP-1 cell differentiation towards foam cells—foam cells being a characteristic feature of atherosclerotic blood vessels. (5) Previously, we found TA-MUC1 downregulation led to a reduction in procoagulant and prothrombotic properties of the cancer cells as well as modulation of the aberrant calcium signalling pathways of cancer cells. Taken together with these current results, this suggests that TA-MUC1 in cancer cells has multiple effects on cholesterol and lipid metabolism, which also impacts other cells in the cellular bioenvironment. TA-MUC1 could thereby act as an important pathogenic effector of atherosclerosis in cancer. These results can also be considered in respect of the therapeutic anti-MUC1 antibody, which was able to reduce the effect of TA-MUC1 on cholesterol metabolism. Modulation of cholesterol metabolism via targeting TA-MUC1 could, therefore, be of great benefit to cancer patients with atherosclerosis.

## 1. Introduction

Atherosclerosis and cancer are chronic diseases considered as two of the main causes of death all over the world. Both diseases arise from multiple factors during different stages in their development but also share common risk factors, comprising genetic alterations, inflammatory processes, uncontrolled cell proliferation, angiogenesis, apoptosis and oxidative stress, during the course of growth dysregulation. This dysregulation involves not only several important molecular pathways but also many aetiological and mechanistic processes from the very early stages of development up to the advanced stages in both pathologies [1,2].

It is well known that cancer may promote atherosclerosis and change the functional and structural markers of subclinical atherosclerosis [3]. Patients with cancer have a high prevalence of atherosclerosis, which may be related to the systemic inflammatory response and impaired vascular endothelial function caused by the mechanism of cancer cell infection and corresponding chemoradiotherapy [3,4]. Despite numerous studies in the field of atherogenesis, there are still great gaps within an already poor understanding of mechanisms that underlie the disease. Inflammation and lipid metabolism violations are undoubtedly the key players; however, whether many other factors, such as oxidative stress, endothelial dysfunction and more especially whether the molecular basis of tumour biochemistry, which causes abnormal lipid metabolism, could also contribute to the pathogenesis of atherosclerosis in cancer patients is still largely unclear.

Mucins comprise a family of heavily glycosylated proteins that are produced in the form of a gel by epithelial tissues in most animals. The most extensively studied transmembrane mucin is MUC1. As a type I transmembrane protein, MUC1 has roles in cell signalling processes, as ligands or receptors and in cellular adhesion. MUC1 is known to interact with multiple signalling cascades affecting the activities of numerous signalling pathways, which involve effects on proliferation, adhesion and apoptosis. An aberrantly glycosylated and overexpressed form of MUC1 occurs in many human epithelial cancers and was termed tumour-associated MUC1 (TA-MUC1) [5]; it is considered as a potential target for cancer therapy. As a multifaceted oncoprotein, TA-MUC1 participates in intracellular signal transduction pathways and regulates the expression of its target genes at both the transcriptional and post-transcriptional levels in cancer cells. Our previous research showed that the level of TA-MUC1 exerts a significant effect on the procoagulant and proinflammatory properties of tumour cells [5]. TA-MUC1 downregulation in breast cancer MCF-7 cells led to reductions in the procoagulant parameters, particularly thrombin generation activity, and, in addition, modulated the aberrant calcium signalling pathways of cancer cells [6].

Altered cholesterol metabolism and regulation of its circulating levels have a major impact on atherosclerosis [7]. It has been shown that lipoproteins and lipid factors, important in the pathophysiology of atherosclerosis, also play a pro-tumourigenic role in several cancers [8]. Lipid metabolism plays a critical role in supporting cancer progression and remodelling the tumour microenvironment. Abnormal blood cholesterol levels are significantly correlated with the risk of different cancers [9]. Targeting the lipid metabolism pathway could provide a novel approach to cancer treatment [10,11]. As a transmembrane protein, TA-MUC1 has roles in cell signalling processes and is overexpressed in tumours of epithelial origin. TA-MUC1 causes aberrant signalling due to loss of apical–basal polarity in cancer. It has been reported that MUC1 regulates the metabolite flux at multiple levels. Serving as a transcriptional co-activator, MUC1 directly regulates expression of metabolic genes [12] and induces alterations in a lipid metabolic gene network [13]. Lack of intestinal MUC1 impairs cholesterol uptake and absorption but not fatty acid uptake in Muc1^−/−^ mice [14]. Therefore, a hypothesis was raised that TA-MUC1, via its effect on intracellular signal transduction pathways in epithelial tissues, could be involved in the modulation of cholesterol and lipid metabolism and contribute to the pathogenetic mechanisms of atherosclerosis-associated cancer.

Foam cells are a type of macrophage that localise to fatty deposits on blood vessel walls where they form through dysregulated lipid metabolism in mammalian macrophages [15]. Excessive accumulation of foam cells leads to necrosis within atherosclerotic plaques and plays a vital role in the initiation and development of atherosclerosis [16,17,18]. The formation of lipid-laden foam cells is a hallmark of both early and late atherosclerotic lesions [19,20]. Foam cells are also associated with chronic inflammation in certain cancers and in metabolic, infectious and autoimmune diseases [15]. It has been widely recognised that monocytes play an integral role in the development and progression of atherosclerosis [21,22]. As a useful model for studying foam cell formation in vitro, THP-1 cell is a human monocytic leukaemia cell line that shares many properties with human monocyte-derived macrophages and has been largely used for studying lipid-laden macrophages that form as a key feature of atherosclerosis [23,24].

This present study was established to investigate whether TA-MUC1 of cancer cells could affect cholesterol and lipid metabolism and whether such an effect was capable of impacting other types of cells within an experimental cellular bioenvironment, especially whether TA-MUC1 could have the potential to exert specific effects on cholesterol plaques forming during the pathogenesis of atherosclerosis. In this investigation, we examined the effect of TA-MUC1 on the cholesterol and fatty acid metabolism of cancer cells via MUC1 gene knock down breast cancer cells. MUC1 inhibitor GO-203 and anti-TA-MUC1 therapeutic antibody also were used within different cancer cells. The technique of coculturing can be used to study cell crosstalk/interaction between two or more different types of cells in a culture plate in vitro model similar to biological tissue [25]. The coculture of monocyte THP-1 cell and breast cancer MCF-7 cells also was performed, in an attempt to find direct evidence that TA-MUC1 could modulate the cholesterol metabolism of other cells. Our results demonstrated that TA-MUC1 was able to modulate the cholesterol metabolism in human breast cancer cells. Reduction or inhibition of TA-MUC1 activity resulted in alteration of the signalling pathways and proteins relevant to cholesterol metabolism. The TA-MUC1 of cancer cells effectively induced monocyte THP-1 cell differentiation towards foam cells. These findings provide additional evidence that TA-MUC1 is an effective modulator for cholesterol and lipid metabolism, which could act as a potential pathogenetic effector of atherosclerosis in cancer.

## 2. Results

### 2.1. TA-MUC1 Modulated the Regulation of Cholesterol and Fatty Acid Metabolism in Breast Cancer Cells

#### 2.1.1. The Effect of TA-MUC1 on Cholesterol and Fatty Acid Metabolism Was Confirmed Using MUC1 Gene Knock down MCF-7 Cells (sh-MUC1-MCF7)

Tumour cells enhance lipid metabolism by increasing exogenous lipid uptake and lipid synthesis, leading to increased intracellular lipid content [10]. Cellular cholesterol metabolism quantitation and cholesterol uptake assays were performed within sh-MUC1-MCF7 cells, using sh-Ctrl-MCF7 and native MCF-7 cells as control. Results show that total cholesterol and free cholesterol were significantly low in sh-MUC1-MCF7 cells compared to sh-Ctrl-MCF7 and native MCF-7 cells (Figure 1A). The cholesterol uptake assay, studying cellular cholesterol trafficking, also shows that the uptake was significantly low in sh-MUC1-MCF7 cells compared to control cells (Figure 1B). This demonstrated that cholesterol metabolism and regulation were modulated by the MUC1 gene in breast cancer cells.

#### 2.1.2. Signalling Pathways and Proteins Implicated in Cholesterol Metabolism Were Also Impacted in TA-MUC1 Cancer Cells

The effect of TA-MUC1 on proteins relevant to cholesterol metabolism and atherosclerosis pathogenesis were also analysed within sh-MUC1-MCF7 cells. The expression levels of several proteins, including leptin (the primary function of which is the regulation of the body fat depot), low-density lipoprotein receptor (LDLR), *ABCA1* (ABC transporter that mediates lipid efflux from macrophages), ACAT1 (which influences the efflux of cellular and lipoprotein-derived cholesterol and causes a pro-atherogenic transformation of macrophages), Krüppel-like factor 4 (KLF4) (which regulates diverse cellular processes as a crucial regulator of foam cell formation) and apolipoprotein B (Apo B) (involved in the metabolism of lipids and is commonly used to detect the relative risk of atherosclerotic cardiovascular disease), were assayed by Western blot in different types of human cancer cell lines, including native MCF-7, shCtrl-MCF7 and shMUC-MCF7, pancreatic cancer cells (PANC-1) and enriched breast cancer stem cells (CSC), while both PANC-1 and CSC cancer cells exhibit a high expression of TA-MUC1 [5]. In addition, the normal human umbilical vein endothelial cell line (HUVEC) was investigated. The results show that the level of MUC1 differed between each of these cell lines (Figure 2A). The levels of other proteins are lower in shMUC-MCF7 cells compared with native MCF-7 and sh-Ctrl MCF7 cells except KLF4 protein. The levels of ACAT1 and Apo B were relatively low in PANC-1 and CSC compared with native MCF7 and the cells derived from MCF-7 (shCtrl- MCF7 and shMUC-MCF7). There may be variations among different types of cancer cells. The levels of all these proteins are relatively low in normal HUVEC cells, which was consistent with its lowest level of MUC1 (Figure 2A). PI3K/*AKT* and RAS/*MAPK* can modulate the activities of various enzymes involved in lipid metabolism in tumours [26]. Our results also show that the activities of p-Akt and p-ERK are lower in HUVEC and sh-MUC-MCF7 cells, both with the lowest level of MUC1 compared to other cells (Figure 2A).

To further confirm the influence of TA-MUC1 on these proteins, normal HUVEC cells were treated with small extracellular vesicles (sEVs) derived from different cancer cells for 3 days and sEVs from a normal epithelial cell line, while MCF12 was used as control. The expression levels of proteins relevant to cholesterol metabolism and atherosclerosis pathogenesis within the sEV-treated cells were analysed by Western blot as well. The results show that the levels of these proteins were able to be induced by sEVs from these cancer cells, along with an enhanced level of MUC1 protein in treated HUVEC cells. It was noticed that the levels of these proteins were relatively low compared to that induced by the sEVs from sh-MUC1-MCF7 cells (Figure 2B). In addition, the activities of p-Akt and p-ERK were also found to be enhanced in HUVEC cells treated with sEVs from MCF-7, sh-Ctrl-MCF7, CSC and PANC cells. The enhancement of the activities of p-Akt and p-ERK was found to be less when treated with sEVs from sh-MUC1-MCF7 and MCF12 cells.

#### 2.1.3. TA-MUC1’s Effect on Cholesterol Metabolism Within Different Cancer Cells Was Further Investigated Using MUC1 Inhibitor GO-203 and Anti-TA-MUC1 Antibody

Mucin 1 (MUC1) is a heterodimeric protein formed by two subunits, N-terminal (MUC1-N) and C-terminal subunit (MUC1-C). As an oncoprotein that forms the cytoplasmic part of the MUC1 transmembrane protein, MUC1-C is overexpressed in many human cancers. It drives cancer growth and progression. Therefore, MUC1-C was chosen as a therapeutic target, with drugs being developed to block its function and inhibit cancer [27]. The MUC1-C inhibitor, GO-203, a cell-penetrating peptide that prevents it from homodimerising and downregulates TIGAR (TP53-induced glycolysis and apoptosis regulator) protein synthesis by inhibiting the PI3K-AKT-S6K1 pathway and, thereby, its oncogenic function, was used to check the effect of MUC1 on the proteins relevant to cellular cholesterol metabolism [28,29].

Three types of cancer cells of MCF-7, CSC and PANC and normal HUVEC cells were treated with the inhibitor overnight, and Western blot assay was performed with the lysate from the treated cells. In comparison to treatment with control peptide, CP2, to non-treated cells (CTL), the results show that the levels of LDLR proteins were reduced in MCF-7, CSC and PANC. The level of leptin was slightly reduced in CSC and PANC cells. The level of Apo B was reduced in CSC cells only (Figure 3A). Also, the levels of these assayed proteins were reduced in normal HUVEC cells.

Considering MUC1-C inhibitor (GO-203) overnight treatment only, sensitivity to variations may also exist among different types of cancer cells. A therapeutic anti-MUC1 antibody—Gatipotuzumab—was also used for this analysis [30]. Three cancer cell lines (MCF-7, CSC and PANC) were treated with Gatipotuzumab for 3 days, then cell samples were harvested for Western blot and cholesterol uptake assays, which were determined as described above. In comparison with both controls (isotype IgG (IgG) control and no antibody (CTL) treated cells), the levels of leptin, LDLR, ACAT1 and Apo- B were reduced in anti-MUC1 antibody (Mab) treated in all three of the cancer cell lines (MCF-7, CSC and PANC). The level of KLF4 was also reduced in MCF-7 and PANC cells except CSC cells, in which KLF4 level was reduced by both the isotype IgG and anti-MUC1 antibody (Figure 3B); however, no significant effect on these proteins was observed in normal HUVEC cells. In addition, the levels of p-ERK and p-Akt were reduced by anti-MUC1 antibody in all three cancer cell lines (Figure 3B).

Cholesterol uptake assay was also undertaken in anti-MUC1 antibody-treated cells. In comparison with both of the controls (CTL and Isotype IgG treated), the levels of cholesterol uptake were significantly reduced by the anti-MUC1 antibody within three cancer cell lines and less reduced in normal HUVEC cells (Figure 3C). All of these results further confirmed a potential effect of TA-MUC1 on cholesterol metabolism.

### 2.2. Cholesterol Metabolism by THP-1 Cells Was Modulated When Cocultured with Breast Cancer MCF-7 Cells

Cell coculture techniques enable the creation of cellular environments in which the physiological and pathological interactions between different cell types can be studied. These interactions can take place by direct contact and by exchange of soluble factors and EVs. Since our research showed that EVs derived from cancer cells had a high level of TA-MUC1 and were able to modulate a number of cholesterol metabolism proteins in normal HUVEC cells, we considered whether TA-MUC1, as a tumour cell-derived effector, also could impact the cholesterol metabolism of other types of cells within cocultured cellular environments. It had been reported that the conditioned medium from the tumour cell lines induced activation and differentiation of THP-1 cells [31]. As a widely useful model for studying foam cell formation in vitro [32], THP-1 cells were used in our cocultured system.

#### 2.2.1. TA-MUC1 Impacted the Cholesterol Uptake of THP1 Cells When Cocultured with Cancer Cells

To make sure THP1 cells could be impacted by TA-MUC1 in a coculture system with cancer cells, NBD-cholesterol uptake assays in cocultured THP-1 cells were performed. Breast cancer native MCF-7, shCtrl-MCF-7 and shMUC1-MCF-7 cells were cultured on the apical part of the insert of a trans-well plate for 2 days, then transferred to 6-well plate also containing THP-1 cells, which were in parallel pre-seeded, and cocultured for 3 days. On the fourth day of coculturing, 20 ug/mL NBD-cholesterol was added to the cell culture for another 24 h. The next day, for the suspension cocultured THP-1 cells, those THP-1 cells that were not observed as differentiated attached cells were collected and counted and equal cell numbers and subjected to the cholesterol uptake assay. It was shown that the cholesterol uptake rates were significantly increased in THP-1 cells cocultured with native MCF-7 and shCtrl-MCF7, but only slightly enhanced when cocultured with shMUC1-MCF7, in comparison with THP-1 only (Ctrl) (Figure 4A). A similar conclusion was reached when the cholesterol uptake assay was performed using flow cytometry (Figure 4B).

#### 2.2.2. TA-MUC1 Induced Pro-Differentiation Proteins in THP-1 Cells Cocultured with Cancer Cells

It is well known that that conditioned medium from tumour cell lines can induce activation and differentiation of THP-1 cells. The changes involved increased expression of macrophage differentiation markers and some of the pro-foam cell formation proteins as well. For example, macrophagic CD146 promotes the formation of macrophage foam cells and retention during atherosclerosis [33]. Low-density lipoprotein receptor (LDLR) on macrophages can facilitate foam cell formation and contribute to the atherosclerotic process under conditions of modestly high cholesterol levels [34]. Expression of very low-density lipoprotein receptor (VLDLR), primarily in macrophages, has been confirmed in human and rabbit atherosclerotic lesions. VLDL receptor pathway involves the foam cell formation mechanism in macrophages [35]. To further determine the effect of TA-MUC1 on THP-1 differentiation in cocultured systems, THP-1 cells were cocultured with native MCF-7, sh-Ctrl-MCF7 and sh-MUC1-MCF-7 for 4 days according to the description above. The cocultured THP cells were subjected to Western blot assay. The results show that the levels of VLDR, CD146 and LDLR were significantly enhanced in the cells cocultured with MCF-7 and sh-Ctrl MCF7 cells, compared with cells cocultured with sh-MUC1-MCF-7 cells and untreated cells (CTL) (Figure 5A). These results indicate that TA-MUC1 is able to modulate THP-1 cell differentiation and induce some of pro-foam cell formation molecules under coculture conditions with cancer cells. As CD146 also can activate numerous signalling pathways and is involved in the activation of NFκB, Erk and Akt signalling pathways [36], *Akt* phosphorylation promotes macrophage foam cell formation [37]. Our data also found that the activities of p-ERK and p-Akt were enhanced in the cells cocultured with MCF-7 and sh-Ctrl MCF7 cells but were at a low level in the cells cocultured with sh-MUC1 MCF-7 cells (Figure 5A).

#### 2.2.3. TA-MUC1 Modulates miRNAs Relevant to the Regulation of Lipid Retention and Differentiation (Foam Cell Formation) in THP-1 Cells Cocultured with Cancer Cells

The extracellular environment can be used alongside genetic modification as an additional tool to control cellular behaviour. It is well known that miRNAs are important regulators of lipid accumulation and monocyte differentiation; significant changes in miRNA expression occur in macrophage “foam” cells, which contribute to macrophage phenotype and inflammation status and also could influence the development of atherosclerotic lesions [38,39]. Several miRNAs relevant to lipid retention and cell differentiation were analysed from the cocultured THP-1 cells. The coculture process used was the same as the description above. Small RNAs isolated from the cocultured THP-1 cells were used to perform miRNA assays. Using the THP 1 cells alone as the baseline of the assay, it was observed that the levels of miR-19b-3p, miR-125b-5p, miR-155-5p and miR-181a-5p, (which increase lipid uptake and favour foam formation) significantly increase in THP-1 cells when cocultured with native MCF-7 and sh-Ctrl-MCF7 compared with THP-1 cells cocultured with sh-MUC1-MCF-7. Two of miRNAs known to reduce lipid accumulation and inhibit foam cell formation were also analysed, of which one, miR-98-5p, only increased in the cells cocultured with sh-MUC1-MCF-7 cells. Another one, miR-21-5p, also increased in the cells cocultured with all three types of cells, but the levels did not achieve significant differences (Figure 5B).

#### 2.2.4. TA-MUC1 Contributes to THP-1 Macrophages Differentiation as Exhibited by Foam Cell Formation When Cocultured with Cancer Cells

As monocytes, THP-1 cells can be stimulated and differentiated as macrophage cells. It had been found that THP-1 cell-derived macrophages cocultured with breast cancer cells (HCC1806) are able to be promoted as tumour-associated macrophages (TAM) [24]. To further determine whether THP-1 cells cocultured with cancer cells can be converted into foam cells, the Oil red O staining assay, which is crucial to examining foam cell formation and to detect lipid droplets distributed in the cytosol macrophages, was used to identify foam cell formation. Since THP-1 cells exhibit a spherical single cell suspension feature, THP-1 cells were treated with PMA (100 ng/mL) for 24 h and induced as THP-1-derived macrophages (M0s), which showed the highest adherence to the culture plate. After that, the PMA medium was removed, THP-1-derived macrophages (M0s) were cocultured with native MCF-7, shCtrl-MCF7 or shMUC1-MCF7 cells pre-seeded in trans-well inserts in PMA-free medium for another 4 days. After that, the cocultured THP-1-derived macrophages in the 6-well plate were stained with Oil red O and observed under light microscopy. Significant Oil red O staining was found in most of the THP-1 macrophages cocultured with MCF-7 and shCtrl-MCF7 cells, but was hardly seen in cells cocultured with shMUC1-MCF7 cells (Figure 6A). This is conclusive evidence that TA-MUC1 of cancer cells contributes to the induction of monocyte (THP-1) cell transition to foam cells in cocultured conditions.

It was also noticed that changes in morphology with increased cell and nucleus sizes were seen in most of the THP-1 macrophages that Phorbol 12-myristate 13-acetate (PMA) induced. Depending on the stimulation factors, M0 macrophages can be polarised into two phenotypes: the classical activated type 1 (M1) phenotype promotes inflammatory responses, while the alternative activated type 2 (M2) phenotype is generally associated with tumour-promoting conditions [40]. To further identify variations in macrophage formation induced by TA-MUC1 in our THP-1 cell coculture conditions, the PMA-treated macrophages were also were treated with one of the bacterial cell wall components—lipopolysaccharide (LPS, 100 ng/mL)—for 4 days, which is widely used to create an inflammatory cell model. The treated cells showed very little Oil red O staining (trace) compared with the macrophages cocultured with cancer cells. Apart from an increase in cell and nucleus size, significant dendrites were also observed to develop around the LPS-treated macrophages (Figure 6B). Western blot assays were also used to check some of the proteins relevant to cholesterol metabolism and foam cell formation with LPS-treated macrophages. The results show that the level of adhesion receptor CD146 increased in LPS-treated cells and was also more significantly enhanced in PMA-treated cells. VLDLR was also induced by LPS treatment, while a reduced LDLR was found in LPS-treated THP-1 cells (Figure 6C). This indicates that differences existed between macrophage differentiation induced by TA-MUC1 of cancer cells and LPS.

## 3. Discussion

Unlike some oncogenes located in specific organs, the overexpression of TA-MUC1 has been found in many types of epithelial cancers involving multiple bio-functions. In cancer cells, TA-MUC1 regulates diverse cellular functions that promote the aggressive and metastatic phenotypes of cancer cells through an intricate interplay of the MUC1-C subunit with various signalling effectors, while also impacting other cells in the cellular bioenvironment, consequently causing a wide range of pathological changes [5,6,41].

It is well known that the coagulation system plays an important role in the development of thrombotic complications in patients with established atherosclerotic disease. Coagulation proteins are not only involved in haemostasis; they also play a role in atherogenesis, particularly in the initial development of atherosclerosis, mainly through protease-activated receptors (PAR)—activation [42]. Thrombin activation of PARs initiates a host of intracellular signalling pathways, including the PI3-AKT/Ca2+/MAPK cascade and subsequent cellular responses [43,44]. Inhibition of coagulation proteins could potentially protect the vessel wall against progression of atherosclerosis [42]. Our previous research showed that a significant reduction in PAR1 and PAR4 coincides with reduced p-Akt, tissue factor (TF) and thrombin activity in MUC1 gene knock down breast cancer cells. Procoagulant proteins, such as FV, FX, FVa, FXa, thrombin and its receptors, PAR1 and PAR4, were also found to be induced by the uptake of tumour-derived EVs containing a high level of TA-MUC1 in normal cell lines [5].

An important element in the atherosclerotic process is vascular calcification, an inappropriate deposition of calcium in arterial wall beds. Calcium mineralisation of the lumen in the atherosclerotic artery promotes and solidifies plaque formation, causing narrowing of the vessel [45]. Atherosclerotic plaque is made up of deposits of fatty substances, cholesterol, cellular waste products, calcium and fibrin. The plaque stability depends on the differential amounts, sizes, shapes and locations of calcification [46]. Our previous research showed that the downregulation of TA-MUC1 is capable of modulating calcium cellular effects via alteration in the expression of calcium signalling proteins and by impacting calcium-induced initiation of procoagulant activity, which is capable of influencing the effect of thrombin on cancer cells as well [6]. Since the atheromatous core is the most thrombogenic component of human atherosclerotic plaques [47], such special effects of TA-MUC1 could impact plaque thrombogenicity and lead to the pathogenesis of atherosclerosis.

It had been reported that MUC1-induced alterations in a lipid metabolic gene network suggests that the role of MUC1 in regulating cholesterol and lipid metabolism could be multifaceted [13]. As more direct evidence, this investigation showed that the reduction or inhibition of TA-MUC1 activity resulted in significant alterations to a number of the signalling pathways and proteins relevant to abnormal cholesterol metabolism—this signature includes proteins for cholesterol and fatty acid synthesis/deposition (e.g., leptin, LDLR, Apo B) and lipid transport (e.g., ABCA1, ACAT1), as well as the crucial regulators of foam cell formation (e.g., KLF4), a consequence of which was that cholesterol metabolism was reduced by reducing or inhibiting the TA-MUC1 gene in breast cancer cells. The TA-MUC1 of cancer cells is also notable in effectively inducing monocytic THP-1 cell differentiation towards foam cells. These findings indicate that TA-MUC1, particularly when aberrantly expressed in cancer, acts as a metabolic master regulator and exerts multiple effects on cholesterol and lipid metabolism, and thereby has the potential to act as an important pathogenic contributor to atherosclerosis.

Based on the conditions of the internal environment, circulating monocytes enter the endothelium and are exposed to a variety of regulatory signals, enabling the cells to rapidly differentiate into tissue macrophages in the vascular intima. The circulating monocytes can be recruited into the tumour microenvironment and converted into tumour-associated macrophages (TAMs). It has been suggested that the TAMs can regulate the survival and proliferation of the surrounding cancer cells and participate in various aspects of tumour formation, progression and metastasis [48]. TAMs also could become polarised into anti-tumoural M1 and pro-tumoural M2 phenotypes as well, both of which can transform into each other with the changes in the internal environment. However, it has been reported that M2-polarised macrophages appear to be the predominant subpopulation, both in atherosclerotic lesions and in malignant tumours.

Lipid metabolism influences macrophage activation, and lipid accumulation confers pathogenic features on macrophages. It has been found that the accumulation of lipids in TAMs can elicit an immunosuppressive macrophage phenotype, which could promote the progression of cancer [49]. During atherosclerosis, dysregulation of lipid metabolic pathways leads to the formation of foam cells in the intima of arteries [18]. Foam cell formation is a key step in the initiation and progression of atherosclerosis [50]. Excessive accumulation of foam cells leads to necrosis within atherosclerotic plaques [51]. Foamy macrophages may be less inflammatory than their non-foamy counterparts [48]. It has even been reported that the TAMs may accumulate lipids; however, their type and their specific roles in tumourigenesis are still considered poorly understood [31].

It had been reported previously that the conditioned medium from the tumour cell lines induces the activation and differentiation of THP-1 cells [52]. This present investigation shows that TA-MUC1 is able to induce THP-1 macrophage differentiation into foam cell formation, while modulating the miRNA and proteins relevant to cholesterol and lipid metabolism as well. However, in Oil Red O staining for THP-1 macrophages induced by lipopolysaccharide (LPS)—M1 macrophages promoted inflammation but significantly less (weak) than those cells cocultured with cancer cells containing TA-MUC1. To try to further find out the difference between the effect of TA-MUC1 and LPS on cancer cells, our other study using an LPS-created inflammatory cell model showed that under LPS treatment, the level of inflammatory factors (TNF-α—M1 macrophage marker) along with several coagulation elements (TF, prothrombin, thrombin and factor X (FX)) were significantly increased in breast cancer cells (sh-Ctrl MCF-7), in contrast to the only slight increase in TA-MUC1 knock down cancer cells (sh-MUC1 MCF7 cell) (Appendix A). All these findings indicate that TA-MUC1 is capable of impacting the LPS effect on the inflammatory cell model and can induce more foam cell formation than LPS; through LPS, it was also able to enhance the level of coagulation proteins. This further confirms that the multiple effects of TA-MUC1 from cancer should be a key element of modulating the cholesterol and lipid metabolism of atherosclerosis pathogenesis in cancer patients.

In summary, atherosclerosis and cancer arise from multiple factors along different stages in their development and also share common risk factors. As a tumour-specific transmembrane glycoprotein, TA-MUC1 is overexpressed within various human carcinomas and has multiple bio-functions, and could cause a wide range of pathological changes. Our previous research had found TA-MUC1 downregulation led to a reduction in the procoagulant parameters and modulated the aberrant calcium signalling pathways of cancer cells. Additionally, this study shows TA-MUC1 has multiple effects on cholesterol and lipid metabolism, which were also able to impact other cells in the cellular bioenvironment. These characteristics indicate that TA-MUC1 could act as an important pathogenic effector of atherosclerosis in cancer. Since coagulation proteins can also play a role in atherogenesis, the inhibition of coagulation proteins could potentially protect the vessel wall against progression of atherosclerosis. Combining antiplatelet and anticoagulant therapy has been demonstrated to improve cardiovascular outcomes in patients with stable atherosclerotic disease [18]. This study also shows that a therapeutic anti-MUC1 antibody was able to reduce the effect of TA-MUC1 on cholesterol metabolism. Therefore, modulating cholesterol metabolism via targeting the multiple effects of TA-MUC1 could be of great benefit to cancer patients with atherosclerosis.

## 4. Materials and Methods

Cell culture and treatment human breast cancer cell line MCF-7 was obtained from the Health Protection Agency culture collection (UK). A human-enriched breast cancer stem cell (CSC) line was purchased from BioMedicure (San Diego, CA, USA). Human pancreatic carcinoma cells lines, PANC-1, human umbilical vein endothelial cells (HUVECs) and THP-1 cells were purchased from ATCC (Manassas, VA, USA). MCF-7, CSC and PANC-1 cells were maintained in DMEM medium supplemented with 10% fetal bovine serum (FCS, Life Technology, Carlsbad, CA, USA), 100 U/mL penicillin and 100 mg/ml streptomycin and cultured in a humidified atmosphere of 5% CO_2_ in air. HUVECs were cultured in human large vessel endothelial cell basal medium with large vessel endothelial supplement (LVES) (Thermofisher, Horsham, UK). PANC-1 cells were maintained in DMEM medium supplemented with 10% FCS. THP-1 cells were maintained in RPMI-1640 Medium with 10% FCS as well. 

sh-MUC1 knock down transfection procedure MCF-7 cells were transfected with MUC1 MISSION shRNA Lentiviral Transduction Particles (Sigma-Aldrich, Gillingham, Dorset, UK) following the manufacturer’s protocol. The stable MUC1 gene knock down cell line (shMUC1-MCF-7) was established and subjected to a series of assays, using MISSION pLKO-1 puro non-mammalian shRNA control plasmid (Sigma)-transfected MCF-7 cells (shCtrl-MCF-7) as control. Puromycin (Sigma-Aldrich, Gillingham, Dorset, UK) was used to select each of the stably transduced cell lines for investigation. 

The cholesterol assay was performed with a Cholesterol Quantification Kit (Sigma-Aldrich, USA) according to the manufacturer’s protocol—the total cholesterol and free cholesterol were measured with the samples of cultured cells. A Cholesterol Uptake Assay kit (NBD-cholesterol uptake, Abcam, Cambridge, UK) was used for the detection of cholesterol taken up by cultured cells. NBD-cholesterol is a synthetic fluorescent probe that enters cells via a different mechanism than lipoproteins, which undergo receptor-mediated endocytosis. NBD-cholesterol’s uptake can be mediated by receptors like SR-B1 for high-density lipoprotein (HDL)-like uptake, or by non-vesicular transport proteins and by other means, depending on its structure and cellular context, allowing researchers to specifically study cholesterol transport pathways [52,53,54,55]. TA-MUC1 acts as a metabolic master regulator in cancer cells by influencing multiple metabolic pathways and not as traditional receptors bind a ligand, leading to intracellular signal transduction. Therefore, NBD-cholesterol uptake was used in this study. During the culture of cells under experimental conditions [52,54,55,56], adherent cell lines were grown in a black clear-bottom 96-well plate, while suspensions of THP-1 cells were grown in 25 cm^2^ flasks or in 6-well plates in coculture experiments. When optimal cell numbers were reached for each application, the cells were washed and incubated within serum and phenol-red free culture medium containing 20 µg/mL NBD-cholesterol for 24 h. After that, the medium was replaced with assay buffer, and the treated cells were analysed using a Spectra Max plate reader (Molecular Devices, Sunnyvale, CA, USA) with filter sets designed for FITC/GFP, or by flow cytometry (CytoFLEX, Beckman Coulter, CA, USA).

For Western blotting, the cells were cultured until 90% confluence in medium containing 10% FBS. At that point, the medium was removed and the cells were washed with PBS. Lysates of harvested cells were analysed by Western blot [57]. Primary antibodies towards MUC1, leptin, low-density lipoprotein receptor (LDLR), ACAT1, KLF4, Apo-B, VLDLR, CD146, ABCB1, ABCA1, ABCG2, ABCC1 and ABCC2 were obtained from Abcam (UK). Primary antibodies within the kits used to analyse the Akt pathway and the MAPK family were obtained from Cell Signaling (Danvers, MA, USA). Densitometry was performed using Invitrogen™iBright™ Imaging Systems (Thermofisher, UK). The bar graphs show the expression and densitometric ratios of each protein relative to GAPDH, AKT or ERK. The mean ratios + S.E. from triplicate identical blots were calculated for each bar.

For anti-MUC1 antibody and MUC1 inhibitor treatment, cultures of human breast cancer MCF-7 and CSC cells, pancreatic cancer PNAC-1 cells and HUVEC cells were each separately treated with a therapeutic anti-MUC1 antibody—Gatipotuzumab, a humanised monoclonal antibody against the TA-MUC1 epitope (MUC1/CD227) (ProteoGenix, Schiltigheim, France)—according to a previously published recommendation of 60–120 µg/mL [30] in Opti-MEM™ I reduced serum medium (Gibco, Waltham, MA, USA) containing 0.2% Lactalbumin Hydrolysate (Sigma, UK) for 3 days. After that time, cell samples were harvested for Western blot and cholesterol uptake analysis as described above. The cells treated with the anti-MUC1 antibody were also treated with MUC1 inhibitor GO-203 (5 μM, Selleckchem, Houston, TX, USA), a cell-penetrating peptide that blocks MUC1-C homodimerisation and, thereby, its oncogenic function [28,58], compared with those treated with the control peptide CP2 (5 μM, Selleckchem, USA) for 48 h. The use of such concentrations was based on a previously published recommendation causing significant inhibition of human breast cancer cells growth but with less impact on cell viability [28,58].

Small extracellular vesicles (sEVs) induced effects on normal human epithelial cells. sEVs were isolated from cell culture media using a Total Exosome Isolation Reagent (Thermo Fisher Scientific, Horsham, UK). Culture medium was harvested from each of several breast and pancreatic cancer cell lines cultured in medium containing 10% exosome-depleted FBS (System Bioscience, Palo Alto, CA, USA) for 2 days. HUVECs were treated with the separated sEVs (5–10 µg of sEV protein concentration/mL or 4 × 10^9^ sEVs/mL, as counted using qNano Gold, IZON Science Europe, Thame, Oxfordshire, UK) within serum-free Opti-MEM™ I reduced serum medium (Gibco™) without phenol red and containing 0.2% Lactalbumin Hydrolysate (LAH, Sigma-Aldrich, Gillingham, Dorset, UK) overnight in 6-well culture plates for 24 h. The sEVs were removed from the treated cells by washing the cells three times with PBS, then the treated cells were used to perform Western blot and miRNAs assays, targeting proteins and miRNAs associated with tumourigenicity and procoagulant properties.

For the miRNA Expression assay, RNA was isolated from cells using an mirVana miRNA Isolation Kit (Life technology, Carlsbad, CA, USA) following the manufacturer’s protocol. RNA concentrations and purity were determined using a NanoDrop spectrophotometer (Thermo-Fisher, Wilmington, DE, USA). The expressions of miRNAs relevant to lipid accumulation and monocyte differentiation were measured using the TaqMan^®^ Advanced miRNA Assays Kit (Applied Biosystems, Altrincham, Cheshire, UK). RUN6B was used as a housekeeping gene miRNA for normalisation. All qPCR reactions were carried out in triplicate with the ABI PRISM 7900HT Sequence Detection System (Life Technology, USA). Relative quantification of the expression levels was determined according to the comparative threshold cycle (ΔCt) method, using untreated cells as controls.

For the coculture experiments, Thermo Scientific™ Nunc™ cell culture inserts for 6-well plates (the porous membrane of the inserts comprised 0.4 µm pore sizes, Thermo Fisher, Wilmington, DE, USA) were used. Breast cancer native MCF-7, shCtrl-MCF-7 and shMUC1-MCF-7 cells were first seeded at a concentration of 1 × 10^5^ cells/per well on the apical part of the insert of a trans-well plate for 2 days, and then replaced with fresh medium, at which point the insert was put in another set of 6-well plates containing human leukaemic cell line THP-1 (ATCC) pre-seeded at 1 × 10^6^ cells/well in Roswell Park Memorial Institute 1640 (RPMI-1640) grown in medium containing 10% fetal bovine serum (FCS, Life Technology, UK), and then cocultured for 4 days. After that period, the cocultured THP-1 cells were used to prepare samples for the analysis of protein and miRNA relevant to lipid accumulation and monocyte differentiation, as well as cholesterol uptake assays.

For lipid (Oil Red O) staining of THP-1 cells, some relevant proteins and miRNAs were modulated in cocultured THP-1 cells after 4 days of the coculture with breast cancer cells, but the monocytic THP-1 cells were not found to be significantly differentiated as attached macrophages. To further determine whether TA-MUC1 in breast cancer cells is able to affect monocyte differentiation and capable of inducing foam cell formation, THP-1 cells were treated with 100 ng/mL phorbol 12-myristate 13-acetate (PMA, Sigma-Aldrich, Gillingham, Dorset, UK), a commonly used reagent to initiate monocyte differentiation. After 24 h, the PMA induced adherent macrophage- THP-1(M0) cells were then washed out of the PMA-containing medium and cocultured with the breast cancer cells on pre-seeded inserts and fresh medium for a further 4 days. After fixation using 4% paraformaldehyde, the cocultured THP-1 cells were stained with the lipid (Oil Red O) staining kit (Sigma-Aldrich, UK), in which cell nuclei were stained with haematoxylin according to the manufacturer’s protocol. Microscopic images were taken using an inverted microscope (Fisher-Scientific, USA) with a GXCAM Premium Microscope Camera (GT Vision, Hagerstown, MD, USA).

## Figures and Tables

**Figure 1 ijms-27-00518-f001:**
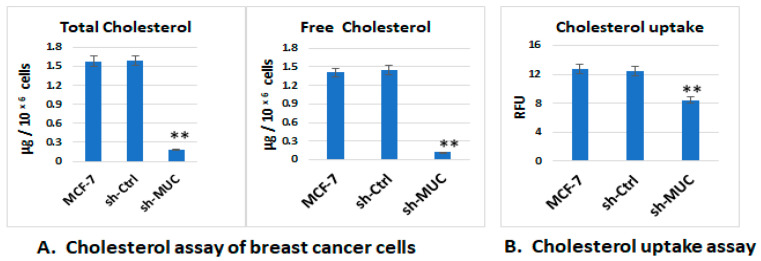
Cholesterol metabolism was reduced in MUC1 gene knock down MCF-7 cells (sh-MUC1-MCF7). (**A**) Cholesterol assay of breast cancer cells. The total cholesterol and free cholesterol were measured with the samples of cultured cells, in which equal numbers of each cell type (1 × 10^6^ cells) were added into 96-well plates and cholesterol quantification assays were performed following the manufacturer’s instructions. The colorimetric-based assay was measured by a microplate reader (570 nm). The cholesterol content was quantified using a cholesterol standard curve. The results show that total cholesterol and free cholesterol are significantly low in sh-MUC1-MCF7 cells. **, indicates a significant difference at *p* < 0.0001, for sh-MUC1 MCF7 cells comparing native MCF-7 and sh-Ctrl MCF7 cells. (**B**) The cholesterol uptake assay. NBD-cholesterol uptake in different cells was measured on a fluorescent plate reader. Results given as relative fluorescence units (RFUs) show that the internalisation of NBD-cholesterol is low in sh-MUC1-MCF7 cells. Data are mean ± SEM of four independent determinations. **, indicates individual *p*-values of Student’s *t*-test for significant differences at *p* < 0.0001 for sh-MUC1 MCF7 cells compared to native MCF-7 and sh-Ctrl MCF7 cells.

**Figure 2 ijms-27-00518-f002:**
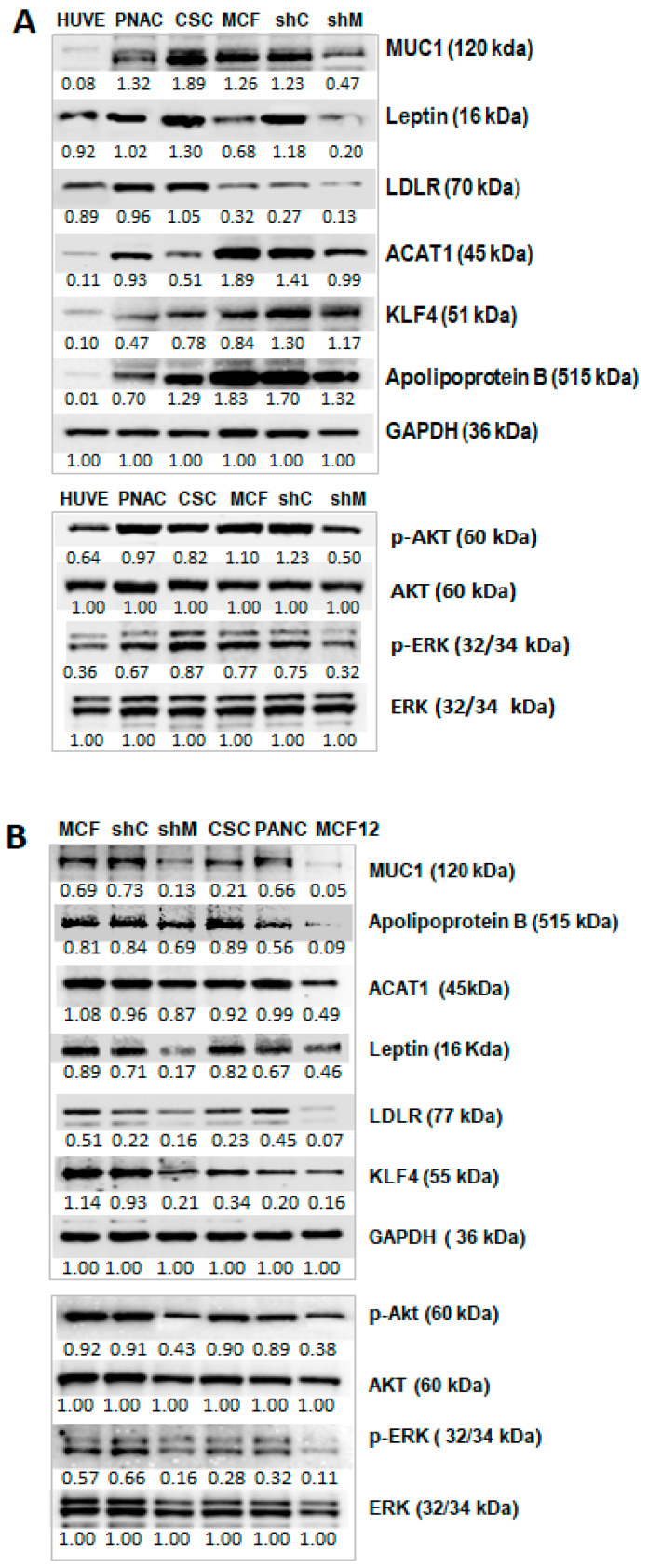
Signalling proteins and proteins involved in cholesterol metabolism and atherosclerosis pathogenesis correspond to the TA-MUC1 levels of cancer cells. Western blot assays were performed to assay the TA-MUC1 effect of cancer cells and sEVs derived from cancer cells. (**A**) The levels of most of these proteins were higher in PANC, CSC, native MCF-7 and shC-MCF7 cells, in comparison with shMUC-MCF7 cells and normal HUVEC cells. (**B**) The levels of these proteins were also liable to induction by sEVs from cancer cells in normal HUVEC cells. Normal HUVEC cells were treated for 24 h with EVs derived from native MCF7, shCtrl-MCF7, shMUC1-MCF7, CSC and PANC-1cells and compared to those treated with EVs from normal MCF12A cells. Results show that the levels of MUC1 and these proteins were induced by tumour-derived EVs in the normal HUVEC line. The number below each Western blot band is the level of expression calculated in respect of the level of GAPDH, ERK or AKT. The density ratio graph of Western blot results is also provided as a Appendix A.

**Figure 3 ijms-27-00518-f003:**
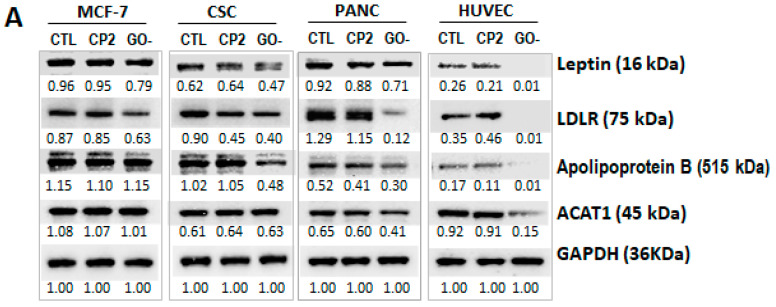
TA-MUC1’s effect on cholesterol metabolism within different cancer cells is liable to restraint by the MUC1 inhibitor GO-203 and by an anti-TA-MUC1 antibody. (**A**) MCF-7, CSC, PANC cancer cells and normal HUVEC cells were treated with MUC1 inhibitor GO-203 (GO) for 48 h and compared to cells treated with a control peptide CP2 and to untreated cells (CTL). Western blot assays of these cell samples show the levels of some proteins involved in cholesterol metabolism and atherosclerosis pathogenesis were reduced. (**B**) The above cells treated with inhibitor GO-203 were also treated with an antibody developed against a tumour-specific MUC1 epitope, Gatipotuzumab (16 µg/mL) (MUC 1Ab), for 3 days. In contrast with an isotype control (Ctrl) and untreated cells (CTL), the anti-MUC1 Ab was able to reduce the levels of most of these proteins in the treated cancer cells. The levels of p-ERK and p-Akt were also reduced by anti-MUC1 antibody in all treated cancer cell lines. The number below each Western blot band is the level of expression calculated in respect of the level of GAPDH, ERK or AKT. The density ratio graph of Western blot results is also provided as a Appendix A. (**C**) Cholesterol uptake assay of anti-MUC1 antibody-treated cells, including native MCF-7, sCtrl-MCF7, sc-MUC-MCF7, CSC, PANC-1 and normal HUVEC cells. In comparison with both of the controls (CTL and Isotype IgG treated), the levels of cholesterol uptake were significantly reduced by the anti-MUC1 antibody within MCF-7, sCtrl-MCF7, CSC and PANC-1 cells, and less reduced in sc-MUC-MCF7 normal and HUVEC cells. Data are mean ± SEM of three independent determinations, * *p* < 0.01 and ** *p* < 0.001 indicate individual *p*-values of Student’s *t*-test for anti-MUC1Ab compared to isotype control (Ctrl) and untreated cells (CTL), respectively.

**Figure 4 ijms-27-00518-f004:**
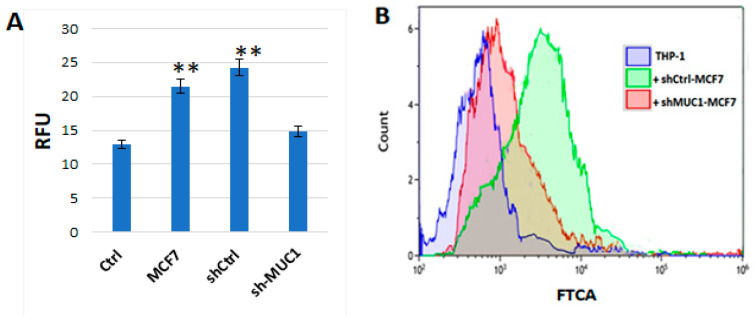
The cholesterol uptake of THP1 cells was impacted when cocultured with TA-MUC1 cancer cells. Breast cancer native MCF-7, shCtrl-MCF-7 and shMUC1-MCF-7 cells were cultured on the apical part of the insert of a trans-well plate for 2 days, then transferred to a 6-well plate containing pre-seeded THP-1 cells and cocultured for 3 days. NBD-cholesterol uptake assays were performed with the cocultured THP-1 cells. (**A**) Cholesterol uptake assay was performed with a fluorescent plate reader, in which equal numbers of each of the experimentally derived cocultured THP-1 cells (1 × 10^6^ cells) were used. In comparison with THP-1 cells only (Ctrl), the results of relative fluorescence units (RFUs) show the internalisation of NBD-cholesterol is low as well in sh-MUC1-MCF7 cells. Data are mean ± SEM of three independent determinations. **, indicate individual *p*-values of Student’s *t*-test for significant differences at *p* < 0.0001 for sh-MUC1 MCF7 cells. (**B**) Cholesterol uptake assay of the cocultured THP-1 cells analysed by flow cytometry (Cyto FLEX-S, Beckman Coulter, Indianapolis, IN, USA). Cholesterol uptake was evaluated in the live cell gate using Kaluza C analysis software Version 2.1. NBD-cholesterol uptake at THP-1 cells cocultured with shCtrl-MCF7 cells significantly (*p* < 0.05) shifted in mean fluorescence as compared to THP-1 cells only and the THP-1 cells cocultured with shMUC-MCF7 cells as well.

**Figure 5 ijms-27-00518-f005:**
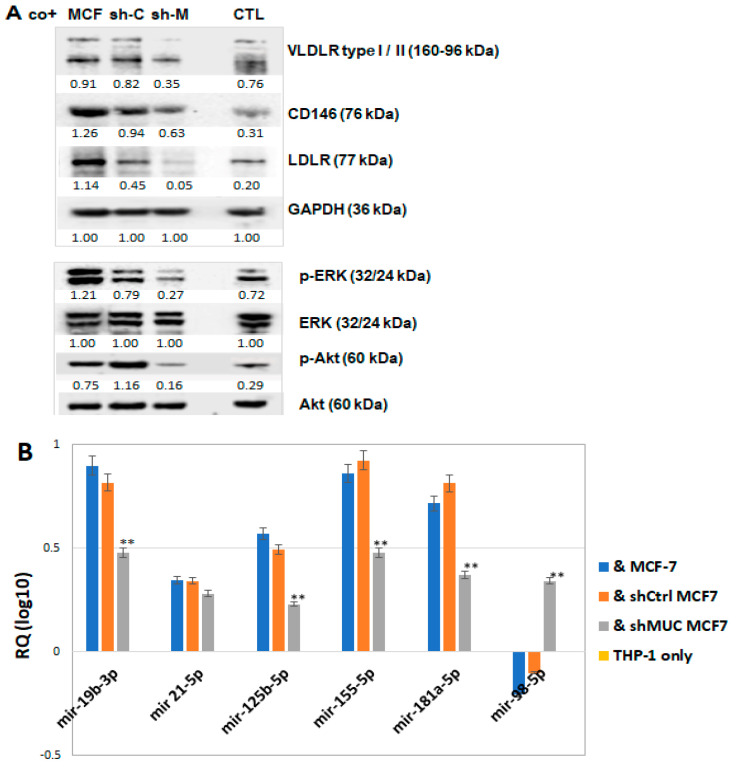
The pro-differentiation properties in THP-1 cells induced when cocultured with TA-MUC1 cancer cells. (**A**) Western blots of THP-1 cells cocultured with MCF-7, shCtrl-MCF7 and shMUC1-MCF7 cells for 5 days, with untreated THP-1 (CTL) as controls. The levels of VLDR, CD146 and LDLR were enhanced in the cells cocultured with MCF-7 and sh-Ctrl MCF7 cells. The number below each Western blot band is the level of expression calculated in respect of the level of GAPDH, ERK or AKT. The density ratio graph of Western blot results is also provided as a Appendix A. (**B**) MicroRNA assay also performed with samples from the above cocultured THP-1 cells. Untreated THP-1 cells only as base line, the expression levels of miRNAs (miR-19b-3b, miR125b-5p, miR155-5p and miR181a-5p), which are associated with macrophage “foam cell” formation and atheroma, were induced by TA-MUC1 cancer cells. Data are mean ± SEM of three independent determinations. ** *p* < 0.001 indicates individual *p*-values of Student’s *t*-test for THP-1 cells cocultured with shMUC1-MCF7 cells compared to native MCF-7 and sh-Ctrl MCF7 cells.

**Figure 6 ijms-27-00518-f006:**
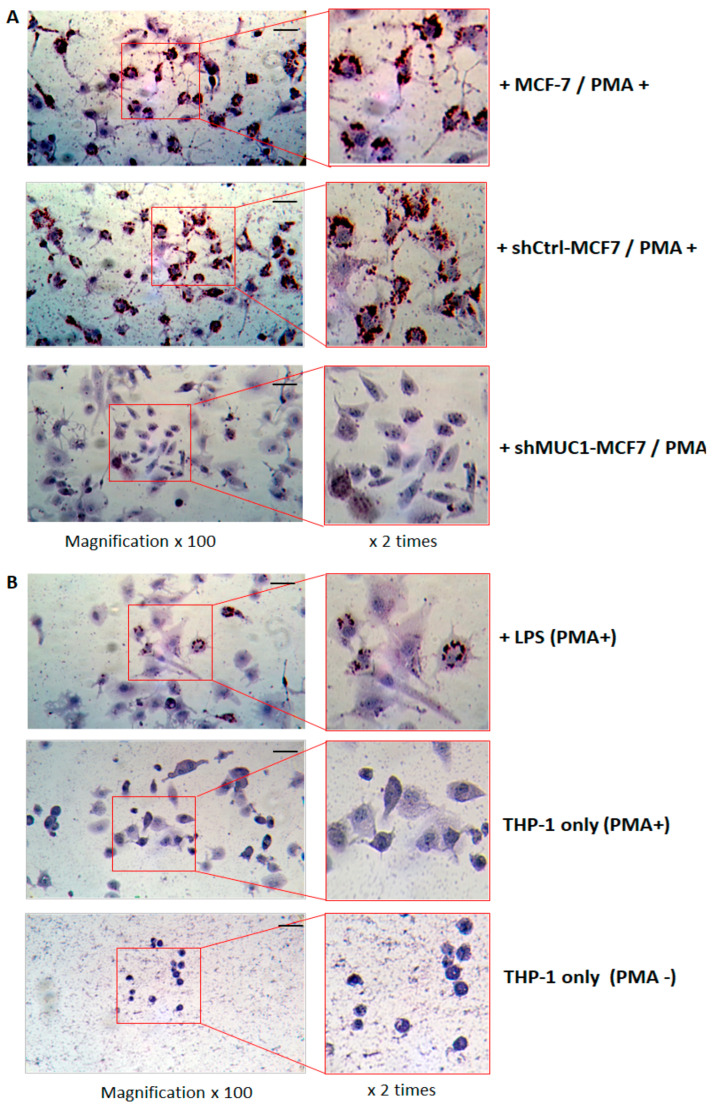
THP-1 macrophages cocultured with TA-MUC1 cancer cells. THP-1 cells were pre-treated with PMA for 24 h, then cocultured with native MCF-7, shCtrl-MCF7 or shMUC1-MCF7 cells for another 4 days. The cocultured THP-1-derived macrophages were stained with Oil red O and observed under light microscopy. Scale bar: 100 µM. Significant Oil red O staining was found in the most of THP-1 macrophages cocultured with MCF-7 and shCtrl-MCF7 cells. (**A**) Less Oil red O staining in the THP-1 cell treated by LPS. (**B**) Western blot assays of the proteins relevant to cholesterol metabolism and foam cell formation with LPS-treated macrophages. (**C**) The number below each Western blot band is the level of expression calculated in respect of the level of GAPDH. The density ratio graph of Western blot results is also provided as a Appendix A.

## Data Availability

The original contributions presented in this study are included in the article/Appendix A. Further inquiries can be directed to the corresponding author.

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
