# Peer review of "Tumour-Associated MUC1 Exerts Multiple Effects on Cholesterol and Lipid Metabolism—A Potential Pathogenic Effector of Atherosclerosis in Cancer"

_ijms, 2026, doi:10.3390/ijms27010518_

Round 1

Reviewer 1 Report (New Reviewer)

Comments and Suggestions for Authors

The authors should work on the following to improve the manuscript 

  1. Abstract : In Background - "Lacking intestinal MUC1 impairs cholesterol uptake" - This statement is unclear , what lacking MUC1 impairs cholestrol uptake? Also a connecting sentence with the previous sentence is necessary 
  2. In the  abstract the results section should be more elaborate and must include statistics.
  3. The hypothesis is not very clear -  is it overexpression of TA-MUC-1 protein that can contribute to the pathogenetic mechnanism of atherosclerosis associated cancer ? or just normal levels of TA-MUC1?
  4. In the introduction, the paragraph on Foam cells and introduction to THP-1 cells, breaks the flow of the introduction. Please revise 
  5. " Our results demonstrated that TA-MUC1 was able to
    modulate the cholesterol metabolism." please clarify which in which cell types are the authors refering to ? Also is it over-expression or normal levels of TA-MUC1 that can modulate cholesterol metabolism?
  6. The statement "Reduction or inhibition of TA-MUC1 activity
    resulted in alteration of the signaling pathways and proteins relevant to the cholesterol metabolism." Can the author clarify what the mean by alteration in signaling pathways and proteins relevant to cholesterol metabolism - does this mean that the signaling pathways are inhibited ? or are they over activated as compared to normal levels ? 
  7. The 'n' values are missing in the results, the authors need to add them to the results sections and also add the p values in the  in the results section 
  8.  Why did authors choose specifically pancreatic cancer cells (PANC-1), and enriched breast cancer stem cells
    (CSC) in addition to the the  MCF-7, shCtrl-MCF7 and shMUC-MCF7 cell lines  for evaluating proteins implicated in cholesterol metabolism?
  9. For all the figures that involve western blot can authors provide statistics  and graphical represenation of the normalized values of all the proteins evaluated ? along with the representative images of western blot analysis 
  10. In section 1.3 the authors talk about MUC1 C , is this an isoform of TA-MUC1, please elaborate further. Also include details on homodimerization is that the mechanism by which the protein gets activated ?
  11. What is the specificity of the inhibitors of MUC1 ? please add refrences talking about the specificity of the inhbitors.
  12. Full form of VLDL should be included in section 2.2 
  13. In section 2.4 - the sentence 'It had been found that THP-1 cell derived microphage cocultured with breast cancer
    cell (HCC1806) are able to be promoted as tumour associated macrophages (TAM)'  - there seems to be a typo - Microphage. Please correct wherever relevant.
  14. The discussion focuses only the foam cell formation, the authors need to discuss all ascpects of thier results here, not just the foam cell ascpect.

Author Response

The authors should work on the following to improve the manuscript

  1. Abstract : In Background - "Lacking intestinal MUC1 impairs cholesterol uptake" - This statement is unclear , what lacking MUC1 impairs cholesterol uptake? Also a connecting sentence with the previous sentence is necessary .

Response -  The suggested amendments have been made. Additional  pertinent information previously  had been described in  more detail in introduction with reference to Ref 14. (Wang H H 2004) ( in page 4 )

  1. In the abstract the results section should be more elaborate and must include statistics.

Response – in our opinion to insert each of the number pertinent parameters would overload the abstract so descriptive words about the system have been added plus the p-value for the entire group ( p< 0.0001) ( in abstract )

  1. The hypothesis is not very clear - is it overexpression of TA-MUC-1 protein that can contribute to the pathogenetic mechanism of atherosclerosis associated cancer ? or just normal levels of TA-MUC1?

Response –This had been described in the introductory section:  An aberrantly glycosylated and overexpressed form of MUC1 occurs in many human epithelial cancers and was termed tumour associated MUC1 (TA-MUC1) [7]. TA-MUC1 means the aberrantly glycosylated and overexpressed form of  of MUC1.  (in  page 3)

  1. In the introduction, the paragraph on Foam cells and introduction to THP-1 cells, breaks the flow of the introduction. Please revise

Response –such amendment has been carried out ( in page 4 and 5 which high lighted )   

  1. " Our results demonstrated that TA-MUC1 was able to

modulate the cholesterol metabolism." please clarify which in which cell types are the authors referring to ? Also is it over-expression or normal levels of TA-MUC1 that can modulate cholesterol metabolism?

Response : Thank you for pointing this  out that the list includes human breast cancer cells ( in page  5).

  1. The statement "Reduction or inhibition of TA-MUC1 activity resulted in alteration of the signaling pathways and proteins relevant to the cholesterol metabolism." Can the author clarify what the mean by alteration in signaling pathways and proteins relevant to cholesterol metabolism - does this mean that the signaling pathways are inhibited ? or are they over activated as compared to normal levels ?

Response: Thank you for pointing this  out  and changes made to describe how  a number of signaling pathways and proteins are relevant to cholesterol metabolism. ( in abstract , and page 12 and 20)

  1. The 'n' values are missing in the results, the authors need to add them to the results sections and also add the p values in the in the results section

Response:  The p values had added in the results section. For “n” values, those for the  cholesterol assay and cholesterol uptake assays have been presented  in figures legends:   Data are mean ± SEM of three or four independent determinations. ( In figures legends)

  1. Why did authors choose specifically pancreatic cancer cells (PANC-1), and enriched breast cancer stem cells (CSC) in addition to the MCF-7, shCtrl-MCF7 and shMUC-MCF7 cell lines for evaluating proteins implicated in cholesterol metabolism?

Response:  Thank you for point out and  has added in manuscript:  thatboth of PANC-1 and CSC cancer cells exhibit  a high level of expression of TA-MUC1 [5] and  were used as control   ( in page 7)  

  1. For all the figures that involve western blot can authors provide statistics and graphical represenation of the normalized values of all the proteins evaluated ? along with the representative images of western blot analysis

Response:  These have been added as the number given below each Western Blot band as the level of expression calculated in respect of the level of GAPDH, ERK or AKT. As there are too many of Western blot band for each figure, it could be too complicated to put an extra bar figure along with each Western blot figure.   

  1. In section 1.3 the authors talk about MUC1 C , is this an isoform of TA-MUC1, please elaborate further. Also include details on homodimerization is that the mechanism by which the protein gets activated ?
  2. What is the specificity of the inhibitors of MUC1 ? please add refrences talking about the specificity of the inhbitors.

Response: following the suggestion this information for question 10 & 11  has been added or  amended in the manuscript ( in page 9)

  1. Full form of VLDL should be included in section 2.2

Response: This has been added ( in page 13)

  1. In section 2.4 - the sentence 'It had been found that THP-1 cell derived microphage cocultured with breast cancer cell (HCC1806) are able to be promoted as tumour associated macrophages (TAM)' - there seems to be a typo - Microphage. Please correct wherever relevant.

Response: These have been amended with a high light indicated   

  1. The discussion focuses only the foam cell formation, the authors need to discuss all aspects of their results here, not just the foam cell aspect.

Response: Thank for suggestion, and we have has added some of our result regarding TA-MUC1  regulating cholesterol and lipid metabolism in the discussion. ( In page 20)  

Reviewer 2 Report (New Reviewer)

Comments and Suggestions for Authors

The authors have explored ways in which Tumor-associated Mucin-1 (TA-MUC1), a transmembrane glycoprotein over-expressed in many epithelial cancers, mediates tumor promotion of atherosclerosis.  They demonstrated that the protein promoted abnormal cholesterol metabolism and monocyte conversion to foam cells.  This study implicates potential therapeutic strategies to ameliorate the cancer-atherosclerosis relationship.  Overall, the ability of this manuscript to convey the sense of the study is hampered by the fact that it is poorly written, revealing a lack of familiarity with the English language.  There are many acronyms that are inadequately defined.  For instance, it is not learned that EVs are extracellular vesicles until the Methods section.  A table of abbreviations would be helpful.  It is difficult to weed through the Western blots to understand effects on protein expression.  Bar graphs paired with them are of different effects.  Bar graphs summarizing the blots (with replicate variance data) should be included.  Does Oil Red O stain blue (Figure 6)?

Comments on the Quality of English Language

The English usage throughout the manuscript is very poor.  As mentioned above, this not only affects the mechanics of the submission, but also the ability to convey the meaning of the study.

Author Response

Thank you for your suggestions. In the initial write up of the manuscript, in which the “Material and Methods” section is located before   Results, following the  format  of IJMS , the some   of the acronyms did not show on the 1st draft appear in this manuscript. Our apology for this, which has now been corrected.

EVs it is shown on page 5 where it first appears.  Other acronyms are now sorted out into  the proper corresponding position.  The manuscript writing also has followed suggestion to do some modification and improvement of the Abstract, Introduction and Discussion,  which are now highlighted and  more detail  are also mentioned and listed in the response to the 1st reviewer 

Oil Red O stain is not blue in colour, it is Haematoxylin which stains cell nuclei blue or purple. It is  a very common Oil Red O staining protocol undertaken according to the manufacturers protocol.    For easy to read, Haematoxylin information has been added in the manuscript.

Round 2

Reviewer 1 Report (New Reviewer)

Comments and Suggestions for Authors

The authors have made additions to the previous version of the manuscript. However, there are more areas where the authors need to improve: 

1) Add the markers for all the western blots to indicate the sizes of the evaluated proteins 

2) Although adding graphs for all the normalized Western blot data can be cumbersome and time-consuming, it is easier to visualize the trends and draw conclusions as a reader. I would recommend that the authors consider adding the graphs for the normalized western blotting data in the supplementary figures 

3) In the methods section, this was observed in multiple places; the cell numbers are indicated as 1 × 106 cells. I would either use a superscript or 10^6 to indicate 1 million cells, etc.

4) The authors need to add the correct punctuation in the manuscript. In the abstract, there needs to be a punctuation between the titles and the content. Also, line 219 starts with italics for no particular reason.

5) In some responses to the reviewers, the authors have mentioned that appropriate changes have been made and added to page 20. There were no highlights made to indicate the changes made, and this is the methods section, so it is unclear why the author has made changes in the methods section 

Comments on the Quality of English Language

The authors need to improve their writing style. It is cumbersome to read the manuscript and understand what the authors want to convey. It is also essential for the authors to use appropriate punctuation and avoid typos. 

Author Response

The authors have made additions to the previous version of the manuscript. However, there are more areas where the authors need to improve: 

  • Add the markers for all the western blots to indicate the sizes of the evaluated proteins 

Response:  This has been added for Fig3 B which missed inclusion of values for AKT and ERK

  • Although adding graphs for all the normalized Western blot data can be cumbersome and time-consuming, it is easier to visualize the trends and draw conclusions as a reader. I would recommend that the authors consider adding the graphs for the normalized western blotting data in the supplementary figures.

Response: The density ratio graph of Western blot data has been provided in the supplementary figures (S1).

  • In the methods section, this was observed in multiple places; the cell numbers are indicated as 1 × 106 cells. I would either use a superscript or 10^6 to indicate 1 million cells, etc.

Response: Thanks for pointing that out. As the most of them were superscript, a few of them that weren’t have been changed to superscript. 

  • The authors need to add the correct punctuation in the manuscript. In the abstract, there needs to be a punctuation between the titles and the content. Also, line 219 starts with italics for no particular reason.

Response: Thank you for pointing out these mistakes, all have been amended. 

  • In some responses to the reviewers, the authors have mentioned that appropriate changes have been made and added to page 20. There were no highlights made to indicate the changes made, and this is the methods section, so it is unclear why the author has made changes in the methods section.

Response: Our last response was based on the initial words file which was returned by IJMS, in which the changes was happened in the discussion section on page 20.  After the editor requested that we follow the IJMS format, change were made and are located on page 16 line 452-464.    

Comments on the Quality of English Language

The authors need to improve their writing style. It is cumbersome to read the manuscript and understand what the authors want to convey. It is also essential for the authors to use appropriate punctuation and avoid typos. 

Response: As a metabolic master regulator in cancer cells, TA-MUC1 exerts its biological effects through multiple aspects, the introduction and descriptions involve multiple fields. Because IJMS is a journal with a broad readership, the background information for different aspects also has to contain more explanation.

Meantime the previous reviewers that have given their opinion have asked us to clarify specifics questions, for example why the NBD cholesterol uptake assay was used in this investigation, therefore in each case we have added a couple of sentences to explain. Probably for this reason, also the descriptions within this manuscript appear lengthy and cumbersome, this was unavoidable.

We have relooked and improved the write -up which we hope will prove to be acceptable.   

Reviewer 2 Report (New Reviewer)

Comments and Suggestions for Authors

There are problems with the English usage in this manuscript, which needs to be reviewed and edited accordingly.

Comments on the Quality of English Language

The English usage throughout the manuscript is very poor.  As mentioned above, this not only affects the mechanics of the submission, but also the ability to convey the meaning of the study.

Author Response

There are problems with the English usage in this manuscript, which needs to be reviewed and edited accordingly.

Response: The manuscript went through again, all errors and mistakes have followed the reviewers and editors suggestions amended.

Thank you for your suggestion.  We have done what we feel is necessary .   

This manuscript is a resubmission of an earlier submission. The following is a list of the peer review reports and author responses from that submission.

Round 1

Reviewer 1 Report

Comments and Suggestions for Authors

The authors presented a very interesting study on how tumor associated Mucin 1 can alter cholesterol metabolism of the tumor cells and other cells nearby. The authors discovered that knocking down MUC1 reduced cholesterol biosynthesis in the tumor cells and nearby macrophages using co-culture experiments. The authors then suggested the potential of MUC1 targeted therapy can help reduce the risk of atherosclerosis in cancer patients. Overall, the study is exciting, but several concerns need to be addressed before publication.

Major:

  1. To ensure statistical validity and reproducibility of the findings, the authors should specify the number of both technical replicates and independent experiments for each result presented. This information should be clearly stated in the methods section and/or the corresponding figure legends.
  2. In Figure 1A, the authors reported OD 570 for their cholesterol assays. It would be more appropriate if the authors could convert the reading to concentration or an absolute value that is normalized to total protein based on their standard curve.
  3. In Figure 1B and 4A, the authors used NBD-cholesterol to monitor how much cholesterol is being taken in by the cells. This kit from abcam measures absorption of unesterified cholesterol by the cells. However, cells rarely absorb unesterified cholesterol in a physiological context because cholesterol is transported by lipoproteins. The authors should try measuring the uptake of fluorescently labeled LDL or oxidized/acetylated LDL instead of NBD-cholesterol.
  4. The authors reported changes in LDLR level in macrophages when they were co-cultured with tumor cells. It would be interesting to also evaluate the changes of scavenger receptor expression level in macrophages when they are cocultured with tumor cells or tumor cells with MUC1 knocked down. Scavenger receptors like CD36, SRA, and LRP1 are responsible for the internalization of oxidized LDL, which are believed to be atherogenic and key for macrophage foam cell formation.
  5. In Figure 6, The authors used oil red o to visualize lipid droplets. The authors should consider using fluorescent lipid droplet stains like bodipy, nile red, or lipidtox instead. This would allow the authors to quantify the results instead of simply report them qualitatively. Also, scale bar information is missing in this figure.

Minor:

  1. Catalogue number information for reagents used in this study is missing.

Author Response

1st. Comments and Suggestions for Authors

The authors presented a very interesting study on how tumor associated Mucin 1 can alter cholesterol metabolism of the tumor cells and other cells nearby. The authors discovered that knocking down MUC1 reduced cholesterol biosynthesis in the tumor cells and nearby macrophages using co-culture experiments. The authors then suggested the potential of MUC1 targeted therapy can help reduce the risk of atherosclerosis in cancer patients. Overall, the study is exciting, but several concerns need to be addressed before publication.

Major:

  1. To ensure statistical validity and reproducibility of the findings, the authors should specify the number of both technical replicates and independent experiments for each result presented. This information should be clearly stated in the methods section and/or the corresponding figure legends.

Response

Thank you for pointing out these errors. We will be more carefully in future.  Changes have been made according to your  suggestions.

  1. In Figure 1A, the authors reported OD 570 for their cholesterol assays. It would be more appropriate if the authors could convert the reading to concentration or an absolute value that is normalized to total protein based on their standard curve.

Response

We prepared a standard curve when we undertook these assays, but noted that  some  published papers also used OD570.    Following your suggestion, the levels are now shown as the  concentration value.

  1. In Figure 1B and 4A, the authors used NBD-cholesterol to monitor how much cholesterol is being taken in by the cells. This kit from abcam measures absorption of unesterified cholesterol by the cells. However, cells rarely absorb unesterified cholesterol in a physiological context because cholesterol is transported by lipoproteins. The authors should try measuring the uptake of fluorescently labeled LDL or oxidized/acetylated LDL instead of NBD-cholesterol.

Response

Cholesterol Uptake Assay Kit (ab236212) which we used provides a convenient tool for studying cellular cholesterol trafficking. The kit employs NBD Cholesterol, a fluorescently-tagged cholesterol, as a probe for the detection of cholesterol taken up by cultured cells.

Before undertaking  this work, we  checked online and AI shown that for a cholesterol uptake assay, NBD-cholesterol is generally considered a better and more versatile option than fluorescently labeled LDL, as it offers greater specificity, flexibility in experimental design, and avoids the complexities associated with labeling and handling whole lipoproteins.

It listed a number of reasons for why NBD-cholesterol is preferred:

  • Specificity and Specificity of Uptake Pathways;
  • Flexibility in Experimental Design;
  • Ease of Use and Handling;
  • Specific Applications- in studies examining cholesterol efflux from cells, intracellular targeting to lipid droplets, and its role in various cellular processes.

It suggested:

If you are interested in the dynamics of free cholesterol uptake and its distribution within the cell, NBD-cholesterol is more suitable. If your focus is on the uptake of whole LDL particles and the role of specific receptors, fluorescently labeled LDL is the better choice. 

NBD-cholesterol can provide detailed information on intracellular localization and trafficking, while fluorescent LDL offers a more direct measure of lipoprotein uptake. 

  1. The authors reported changes in LDLR level in macrophages when they were co-cultured with tumor cells. It would be interesting to also evaluate the changes of scavenger receptor expression level in macrophages when they are cocultured with tumor cells or tumor cells with MUC1 knocked down. Scavenger receptors like CD36, SRA, and LRP1 are responsible for the internalization of oxidized LDL, which are believed to be atherogenic and key for macrophage foam cell formation.

Response:

This  is a good suggestion.  Since CD36, Scavenger Receptor A (SRA), and Low-Density Lipoprotein Receptor-Related Protein 1 (LRP1) are all scavenger receptors that play crucial roles in cellular processes, particularly in lipid metabolism, inflammation, and immunity, with their expression and function often being dysregulated in various pathological conditions like atherosclerosis and cancer.  The microRNAs assay of this investigation also found a MUC1 effect on miR-155-5p which is implicated in atherosclerosis and foam cell formation, and often involves CD36.

As a transmembrane glycoprotein, TA-MUC1 is overexpressed in many types of epithelial cancers and therefore capable ofimpacting a number of different proteins and involve multiple effects.  Ours investigation showedthat several proteins implicated in cholesterol metabolism were impacted in TA-MUC1 cancer cells.  Examples were leptin, LDLR, apolipoprotein B (Apo B), ABCA1, ACAT1 and Krüppel-like factor 4 (KLF4), the most of which mediates the regulation of fat depot lipid efflux from macrophages as a crucial regulator of foam cell formation and causes a pro-atherogenic transformation of macrophages. CD146, Low density liproprotein receptor (LDLR) and VLDL receptor, which promotes / facilitate the formation of macrophage foam cells as were also be assayed in this investigation. 

We will take your advice to investigate CD36, SRA, and LRP1 in our further work.

  1. In Figure 6, The authors used oil red o to visualize lipid droplets. The authors should consider using fluorescent lipid droplet stains like bodipy, nile red, or lipidtox instead. This would allow the authors to quantify the results instead of simply report them qualitatively. Also, scale bar information is missing in this figure.

Response

Thank you again for your advice.

Oil Red O and fluorescent stains like BODIPY, Nile Red, and LipidTox are used to visualize lipid droplets, with each offering distinct advantages and disadvantages: Oil Red O is a well-known lipid dye with drawbacks including arbitrary quantification and the need for fresh preparation, while fluorescent dyes like BODIPY offer excellent photostability and spectral versatility, Nile Red provides high sensitivity and is suitable for flow cytometry, and LipidTox is known for its high affinity for neutral lipid droplets and compatibility with high-content screening.

We have found that even Oil Red O stain is well-established as a method for visualizing cytoplasmic lipid content, but has disadvantages in that the method is sensitive to preparation conditions, requiring fresh working solutions. For future work we will need to consider to use fluorescent dyes.  

Scale bar is added in this figure. Thank you

Minor:  Catalogue number information for reagents used in this study is missing

Response   This information has now been added

  • nd Comments and Suggestions for Authors

It is an interesting and novel article. At the cellular level, the authors' research found that the overexpression of TA - MUC1 in cancer has an impact on cholesterol and lipid metabolism, which is a potential pathogenic effector of atherosclerosis. The methods in this paper are correct. The presentation is clear. The research results are sufficient to draw the conclusion that regulating cholesterol metabolism through targeting the multiple effects of TA - MUC1 is of great benefit to atherosclerosis in cancer patients. However, there are some minor revisions as follows:

  1. The abbreviations in the chart should have corresponding full names. For example: P4 Figure1 sh-MUC1: MUC1 gene knockdown MCF - 7 cell
  2. P5 The seventh line from the bottom should start a new line. That is 1.3 TA-MUC1........

P8 The last line of the second last paragraph should strat a new line. That is 2.2 TA- MUC1

Response

All of above errors have been amended. Thank you 

  1. P6 The third line: the levlels of all of these proteins were reduced in normal HUVEC cell  do not match figure3A. The levels of GAPDH were same in nomal HUVEC cell.

Response

GAPDH has been used in the present study as a house keeping gene, to show protein samples have been equally loaded.  Our description may not be not rigorous enough, so it has changed as:   the levels of these assayed proteins were reduced in normal HUVEC cells.

Reviewer 2 Report

Comments and Suggestions for Authors

It is an interesting and novel article. At the cellular level, the authors' research found that the overexpression of TA - MUC1 in cancer has an impact on cholesterol and lipid metabolism, which is a potential pathogenic effector of atherosclerosis. The methods in this paper are correct. The presentation is clear. The research results are sufficient to draw the conclusion that regulating cholesterol metabolism through targeting the multiple effects of TA - MUC1 is of great benefit to atherosclerosis in cancer patients. However, there are some minor revisions as follows:

1 The abbreviations in the chart should have corresponding full names. For example: P4 Figure1 sh-MUC1: MUC1 gene knockdown MCF - 7 cell

2 P5 The seventh line from the bottom should start a new line. That is 1.3 TA-MUC1........

P8 The last line of the second last paragraph should strat a new line. That is 2.2 TA- MUC1

3 P6 The third line: the levlels of all of these proteins were reduced in normal HUVEC cell  do not match figure3A. The levels of GAPDH were same in nomal HUVEC cell.

Author Response

  • 2nd Comments and Suggestions for Authors

It is an interesting and novel article. At the cellular level, the authors' research found that the overexpression of TA - MUC1 in cancer has an impact on cholesterol and lipid metabolism, which is a potential pathogenic effector of atherosclerosis. The methods in this paper are correct. The presentation is clear. The research results are sufficient to draw the conclusion that regulating cholesterol metabolism through targeting the multiple effects of TA - MUC1 is of great benefit to atherosclerosis in cancer patients. However, there are some minor revisions as follows:

  1. The abbreviations in the chart should have corresponding full names. For example: P4 Figure1 sh-MUC1: MUC1 gene knockdown MCF - 7 cell
  2. P5 The seventh line from the bottom should start a new line. That is 1.3 TA-MUC1........

P8 The last line of the second last paragraph should strat a new line. That is 2.2 TA- MUC1

Response

All of above errors have been amended. Thank you 

  1. P6 The third line: the levlels of all of these proteins were reduced in normal HUVEC cell  do not match figure3A. The levels of GAPDH were same in nomal HUVEC cell.

Response

GAPDH has been used in the present study as a house keeping gene, to show protein samples have been equally loaded.  Our description may not be not rigorous enough, so it has changed as:   the levels of these assayed proteins were reduced in normal HUVEC cells.

Round 2

Reviewer 1 Report

Comments and Suggestions for Authors

I have a serious concern about the design of NBD-cholesterol experiment and the scientific question it aims to answer.

As I commented earlier, cells rarely absorb unesterified cholesterol in a physiological context because cholesterol is transported by lipoproteins. Measuring how much cells absorb free cholesterol has no physiological importance and cannot support the authors' scientific narrative presented in this manuscript.

Furthermore, NBD-cholesterol is a poor mimic of cholesterol. It is polar and water soluble because of the addition of NBD group. It has altered membrane behavior, trafficking, and metabolic processing compared to cholesterol. Other fluorescent cholesterol analogues like bodipy-cholesterol, DHE, or CTL, having their own drawbacks, are at least hydrophobic and behave closer to cholesterol than NBD-cholesterol (Reviewed in Solanko et al., Lipid Insights 2016). Nevertheless, free cholesterol uptake is not an experiment that will support the authors' narrative as I pointed out earlier.

I regret to recommend rejection of this revised manuscript. Despite the authors' revisions, a fundamental flaw noted in the initial review remains unaddressed. I don't believe the manuscript's claim on cholesterol uptake was properly validated, and a suitable design was not presented in the revision. Without this critical evidence, the scientific narrative of the manuscript is flawed. Therefore, I cannot recommend this manuscript for publication.